# Reflections on the Impact of an Intergenerational Digital Storytelling Program on Changing Attitudes and Fostering Dialogue and Understanding across the Generations

Mark Silver *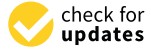 and Lysha Zhi Yan Lee *

Department of Psychological Sciences, School of Health Sciences, Swinburne University of Technology, John St., Hawthorn 3122, Australia
* Correspondence: msilver@swin.edu.au (M.S.); lzlee@swin.edu.au (L.Z.Y.L.)

**Abstract:** Digital storytelling (DST) has the primary goal of giving underrepresented voices a platform to be seen and heard. Adding an intergenerational dimension can bring about many other benefits for all participants as well as the wider community. This article presents a reflection on the Positive Ageing Digital Storytelling Intergenerational Program (PADSIP), outlining the various elements involve in program planning and implementation, reflecting on the past 15 years of program delivery, and underscoring future directions. PADSIP brings together older adults from both community and residential care settings with high school students in an intergenerational context. The process involves collaboratively creating digital stories that explore shared passions and lived experiences. Over the past 15 years, the program has evolved to include various adaptations to accommodate diverse groups, including neurodiverse individuals and those with disabilities. The program, originally taking a ten-session weekly format, has even become an integral part of school curricula in one local high school. Although the COVID-19 pandemic prompted adjustments including temporary transition of program meetings to online platforms as well as video production assistance, the intergenerational bonds and meaningful dialogues remained strong. By challenging stereotypes and fostering deeper connections, the program highlights the potential for intergenerational DST to positively reshape attitudes and understanding among participants. Current and future program research seek to delve into the mechanisms that facilitate such transformative outcomes, investigating the in-depth connections and communication that characterise the intergenerational DST approach.

**Keywords:** intergenerational; digital storytelling; PADSIP (Positive Ageing Digital Storytelling Intergenerational program); community engagement; international digital storytelling conference 2023

## 1. Introduction

### 1.1. Digital Storytelling

Digital storytelling (DST) is a multimedia narrative process that uses technology to convey a story (Robin 2015), combining images (still and moving), sound, music, text, and narration to create a short film documenting lived experiences (Lambert 2010). As a genre, DST has had socio-political roots being and recognised as a source of empowerment by providing marginalised and powerless groups with a "voice" (Bhar et al. 2019). DST is increasingly being used around the world for educational purposes (Robin 2015), leaving legacies (Hausknecht et al. 2019), as well as being a medium to explore oral histories and preserving cultural knowledge (Cunsolo Willox et al. 2013).

The use of DST to improve the health of older adults is an emerging area of research, with existing studies suggesting that DST may be used with older adults as a tool to improve mood, enhance memory (Woods and Subramaniam 2016), increase social connectedness (Hausknecht et al. 2019; Stenhouse et al. 2013), encourage personalised care practices among those who require it (Stenhouse et al. 2013), and promote intergenerational learning (Hewson et al. 2015; Loe 2013).

*1.2. Intergenerational Digital Storytelling*

Whilst the research in the areas of DST and intergenerational programs are independently growing, intergenerational DST as a research area remains relatively new. Benefits of intergenerational storytelling for both cohorts include reducing generational stereotypes, promoting communication and respect, and diminishing social barriers (Hewson et al. 2015; Charise et al. 2022). The understanding across generations that such programs can generate may also go a long way in finding creative solutions to the issues that we all face today. DST, at its core, provides a forum for people who are disenfranchised and feel unheard and invisible to share their message through an enabled voice.

Including an intergenerational dimension offers extra opportunities to cultivate mutually beneficial relationships and communication across generations. This approach can enhance community connectedness and cohesion. As defined by the International Consortium for Intergenerational Programs, intergenerational programs are "social vehicles that create purposeful and ongoing exchange of resources and learning among older and younger generations" (1999). Furthermore, they develop relationships and generate conversations across generations that are mutually beneficial, building a stronger sense of community cohesion. They provide an important avenue for an exchange of talents, resources, knowledge, wisdom, and experience that fosters hope and resilience, as well as challenge ageism and stereotyping. Fostering understanding and learning between the generations can maximise meaningful connections and communication, building relationships and strengthening identities.

## 2. Program Overview

### 2.1. History and Context

This article offers a reflection on an intergenerational DST program, "Positive Ageing Digital Storytelling Intergenerational Program (PADSIP)", reporting its various elements and observed impacts, as well as the future directions. PADSIP started in Melbourne, Australia, in 2007. It was developed as part of Swinburne University's Wellbeing Clinic for Older Adults as an adjunct to the Clinic's Counselling and Support Program, seeking to address some of the underlying factors impacting on older adults' emotional wellbeing.

Since then, the program has been delivered in more than nine high schools with students from Years 7 to 10 meeting weekly with older adults living in both the community and in residential care.

### 2.2. Program Aims

PADSIP's mission is to build transformative connections across generations, forging empathy, trust, and mutual understanding through the power of storytelling and co-creation. We create an inclusive and respectful space where seniors, volunteers, students, and organisers build meaningful relationships, ensuring all voices are valued. Digital stories are used to capture the multigenerational process and the histories of participants. These stories aim to create transformative and sustained impacts for participants and the broader community.

As a social intervention, the program aims to improve the psychosocial health and wellbeing of all participants, including mood, life satisfaction, sense of identity, and social connectedness.

### 2.3. Theoretical Frameworks

There are number of theories that underpin the program and contribute to the transformative nature of PADSIP. Narrative therapy (Cowley and Springen 1995) and reminiscence practice (Butler 1963, 1974, 1980) form the basis of the storytelling approach. In a trusting atmosphere, when individuals tell their stories and receive rich acknowledgment, they share experiences and emotions, finding common ground. This process leads to a breakdown of feelings of isolation and difference, setting the scene for a more nurturing environment. Group work theory (Phillips and Phillips 1993) enables us to comprehend

how mutual support generates its own power, fostering a positive caring atmosphere. This atmosphere empowers group members, giving them agency and more control over their lives. Trauma-informed practice, discussed in the literature with the core characteristics of trust, safety, choice, collaboration, and empowerment (Harris and Fallot 2001; Knight 2019), helps us find ways to maintain an emotional safe space where vulnerabilities can be shared and deeper connections made. Therapeutically informed practice (Monson et al. 2022) links to the transformative or "healing" elements of such programs where identity, meaning and purpose feature strongly. There is a strong influence here also of existential theory (Whittaker 1974; Turner 1974), where reflection and meaningful conversation has its important place.

The effects of non-familial intergenerational practice and programs have been attributed to several theories. In particular, intergroup contact theory (Allport et al. 1954; Pettigrew 1998), an established concept in social psychology studied widely as a vehicle for promoting cooperation and reducing conflict and prejudice, is cited widely. This effect has been established across 1164 different studies of intergroup relations between various types of groups (Pettigrew and Tropp 2006). Specifically, the most powerful type of contact with the highest potential to improve attitude is direct contact via intergenerational friendship (Drury et al. 2017). On top of improving young people's attitudes towards aging, intergenerational friendships have also been associated with positive perceptions of providing care for older people, as well as an increased interest in studying aging. Having at least one older friend can decrease stereotyping of older adults (Drury et al. 2017). Intergenerational contact through a program like PADSIP is thus theorised to reduce ageism and increase cohesion amongst generations.

### 2.4. Program Elements

In the following sections, the various elements that go into the planning and implementation of PADSIP are discussed in a process-oriented approach.

#### 2.4.1. People and Participants

A PADSIP team, including program managers, a videographer, volunteers, a research manager, and an advisory body, oversee the planning process and liaise amongst stakeholders.

The planning stage involves several factors. First, the type and size of the program is considered. A decision about how many students, older adults, volunteers, and other stakeholders is made, paying attention to balancing the right mix of the different elements and characteristics of the group as a whole. This includes being aware of stakeholders' needs and interests (e.g., understanding the constraints of the location where participants will meet).

To recruit participants, the team considers and decides the required number of roles, in terms of volunteers, older adults, and young people/students. It is crucial for roles and expectations to be communicated as clearly as possible. For example, the volunteers can take on the role of being either/or both facilitators and story collectors. We have found that the most effective way of recruiting is by a personal approach either by phone or face to face meetings. Publicising the program through a flier, email, and social media can be best viewed as an adjunct rather than the first port of call.

PADSIP is in every way a participant-centred program. It is thus essential in the planning and execution process to be as fully aware as possible of the unique world in which participants live in. It is not only about considering cognitive, learning, and emotional issues, but also about what might present as difficulties for participants in the PADSIP environment (e.g., needing support staff to facilitate communication, needing breaks). Organisers need to consider factors including whether older adults live in residential facilities or in the community, or what age the students are, and above all, the particular interests and needs of all those involved in the program.

There has also been an evolving role for our volunteers. Each program invites volunteers from community groups or tertiary students to assist with the program. Volunteers are key to facilitating interaction in group discussions, especially where there may be awkwardness present or lack of confidence from students or adults, but also physical and cognitive challenges (e.g., hearing impairments). Sometimes participants may also have difficulty in expressing their views. Volunteers can facilitate communication and can help to ensure that the older adults and students are front and centre in the program. As an example, in a program where there were students with mild intellectual disability and older adults with varying levels of cognitive or physical disabilities, we invited a volunteer to join each resident–student pair during their conversations. The volunteer facilitated the interaction by, for example, prompting the use of a question sheet, making suggestions to either participant, or offering follow-up questions when there were silences.

Our volunteers have reported that they want to strengthen their connection with community; to feel a sense of contribution; build self-esteem and confidence; and further develop communication, language, and engagement skills of being a facilitator. As the program has evolved, the needs of the volunteers have become another factor to consider in the planning and running of the program.

### 2.4.2. Structure and Activities

Over the years, we have learnt that storytelling provides an effective means of facilitating meaningful interaction. In particular, DST allows for both the process involved in creating the stories together, as well as the product created, to be therapeutic and transformative. The power in uncovering a voice to speak and be an advocate for change is often underestimated. We need to provide a voice for the voiceless, disenfranchised, and ignored. The genre of DST is achievable within limited resources and time frames. The technical aspects equalise mutual learning, especially for the students, allowing them a teaching role. There is an emphasis on creativity, versatility, and adaptability with a treasured record to share with a wider community.

In terms of the program structure, we need to consider how many weeks the program should run for, where to meet (i.e., school, aged care facility, both or other community setting), and dates for the Information Launch and Presentation events (discussed below).

Decisions also need to be made about who will do the filming and editing (the students, adults, videographer, or a mixture). In our experience, we often have a documentary logging the progress of the program as it develops, which can let family members and the school community keep engaged with what is happening in the program.

It is important that the production phase still allows for maximising time for interaction and discussion amongst participants. Keeping to a time frame through the different phases (introductions, planning, interviewing, scripting, filming, editing, post-production) enables a quality finished product which honours the dignity of those involved. It is essential that the participants feel a sense of ownership over the finished digital product. The different uses of the digital stories produced can also add a positive dimension to the program. They can be shown to family, friends and staff with pride but also be a lasting legacy for future generations to enjoy.

At commencement, an information launch is held, where the program is explained to participants or those intending to join. A celebratory presentation is also organised at the end of the program, showcasing what has been achieved and celebrating relationships fostered. These are community events where not only the families and friends of participants are invited, but also members of the broader community. These events are a way of building as a sense of community as well as generating a very special atmosphere. They can in essence be described as "definitional ceremonies" (Strauven 2016), engendered with meaning where certificates are presented and participants are richly acknowledged in front of their local community.

### 2.4.3. Logistics

Logistic issues are always paramount to the success of any program, including details of aspects such as food, transportation, and equipment needed (e.g., mobile phones, tablets, microphones, cameras). Sometimes we need to use a shared drive editing system (e.g., WeVideo) so that projects can be accessed by all those involved.

Our experience has been that pre-program preparation can be very valuable. Meeting individuals involved in each group (e.g., volunteers, adults, youth, staff) to address practical and administrative issues, including legislative requirements (e.g., Working with Children's checks, police checks, consent documents) as well as safety requirements including masks, Occupational Health and Safety issues, and risk assessments (e.g., first aid, disability access, toilets) are all worthwhile investments to pursue.

### 2.4.4. Training and Induction

Training is a major part of the preparation involved to set all stakeholders—including the young people, older adults, volunteers, educators, aged care staff, and organisers—up for success. This involves running separate induction sessions for participants and volunteers to understand the program and its aims, as well as to discuss their roles and expectations around involvement. In addition, we invite them to share their background, skills, knowledge, and motivations for getting involved.

Education about the worlds of youth and adults is also key. We have found that education, particularly around topics they may not be typically exposed to, such as disability (in both younger and older cohorts), helps to prepare stakeholders for what to expect and inform appropriate behaviour. This may range from introducing the setting that aged care residents live in to the unique characteristics of neurodivergent individuals. These sessions also act as an opportunity for participants to ask questions and clarify doubts. Such preparation has been highlighted in the literature by various programs to be an important precursor to successful intergenerational interactions (Rice and Chandler 2019; Chung 2009).

### 2.5. *Implementation Challenges*

In terms of implementation, the challenge for PADSIP has been to develop activities that bring out deeper connections, facilitate meaningful conversations, and foster greater understanding amongst the participants. There are many questions around what activities work best and why; where do the "stories", especially digital stories, fit in these activities; and where does the filming of both the stories themselves and any documenting of the program fit in.

A priority for PADSIP has been that the process be just as important as the product. It has always been important to maximise the interaction time amongst participants, emphasising understanding, communication, and connection. The challenge here is to somehow balance planning and structure with a free-flowing nature, which allows for spontaneity and creativity to naturally emerge.

In terms of policies and procedures, it is important to be aware of such areas as Occupational Health and Safety (OHS), insurance and liability, and role descriptions. Participants have to acknowledge and understand the relevant policies, including Police and Working with Children's Checks, as well as COVID restrictions and mandates, as applicable.

Enabling these programs to get started has been a major learning experience in itself. The initial idea for a program can come from an older adult group, a youth group or school, or a community group such as a local council or non-for-profit charity. It is important to find an appropriate partner with consideration given to the goals and aims of the program. There needs to be a good understanding of the different problems that need to be solved by each participant group and how these intersect. Another important step is to find funding; look at the budget, in-kind resources available, scope, and scale of the program; and then to identify the roles needed. We have found that setting up an advisory group and building

a team of committed members has been extremely valuable. They can assist greatly in developing aims and goals, choosing from a range of possibilities such as relevance to the school curriculum and identifying and meeting the needs of participants and stakeholders.

*2.6. Adaptability and Diversity of Programs*

A typical PADSIP program involves students meeting with older adults living both in residential care and in the community. We have discovered over the years that we have had to find ways of adapting the program to meet the needs of various stakeholders and contexts. Table 1 describes a range of PADSIP programs across the years.

**Table 1.** Description of various PADSIP programs conducted since 2007.

| Students | Age(s) | Elders | Year(s) |
| --- | --- | --- | --- |
| Auburn High | Years 7 to 9 | Uniting Agewell (UA) Elgin St Centre, UA Tandera and Condare Court, Mecwacare Trescowthick Centre, City of Boroondara community | 2007–present |
| Forest Hill Primary | Year 6 | UA Strathdon | 2010 |
| Elwood High | Year 8 | UA Girraween | 2011 |
| Fernhill High | Year 8 | U3A Knox | 2018 |
| Camberwell High | Year 9 | Boroondara Stroke Support Group, City of Boroondara community | 2019 & 2022 |
| Sandringham College | Year 10 & 12 (VCAL) | Fairview Aged Care, City of Bayside Community | 2019 & 2022 |
| Kids Like Us/2eHub (Neuro-diverse) | Ages 8 to 18 | Sandy Beach Community Centre, Sandringham Lions Club | 2022 |
| Ashwood School (Intellectual Disability) | Years 11 to 12 | OPAL Waverley Valley Aged Care, Monash Lions Club, Power Neighbourhood House | 2022–2023 |
| Yackandandah Primary | Year 6 | Yackandandah Health and Local Community | 2023 |
| Sandringham College | Year 12 (VCAL) | Sandy Beach Community Centre | 2023 |

During the COVID-19 pandemic and social isolation measures implemented, PADSIP was successfully adapted to an online virtual format. The general structure and focus of the program were maintained, with participants virtually meeting and interacting on platforms such as Cisco Webex. Virtual features such as breakout rooms were utilised to allow participants to converse in smaller groups. Staff and student volunteers in aged care facilities supported participating residents with the technology, and program organisers and other volunteers facilitated conversations in these "rooms". Working within the limitations of pandemic restrictions, the PADSIP team had to direct and take on the production of the videos, yet we were careful to still maintain full student direction, maximising the input and agency of all the participants.

The program has also been adapted to students with disabilities. One program was developed with neurodiverse students and another program adapted to students with mild intellectual disability.

We have also found that a significant number of students in mainstream schools have learning/emotional issues and that this program can assist with their engagement with learning at school.

Many of the older adults involved also have a range of significant physical and cognitive disabilities, and this program has assisted with their feeling a part of a community.

Additionally, there is a lot of potential in embedding this program within the education system so as to achieve sustainability. As an example, in 2022, a ten-session weekly program was included as an integral part of the Year 9 curriculum at one school, and in 2023, it became integral to a Year 12 curriculum at another school in Victoria, Australia. In terms of scalability, this program has also spread to other parts of Australia with larger-scale programs—for example, in 2022 a PADSIP program in Queensland involved 120 Year 7 students from one school and 35 older adults.

## 3. Evaluation Methods

These programs have anecdotally and observationally made significant impacts, not only on an individual level for participants, but also for the broader community. Some examples of these include a school adopting and offering PADSIP as an elective subject; improved relationships between staff and residents in aged care settings; and local community involvement through events such as screenings of digital stories made by students and older adult participants in local cinemas. We would like to examine whether PADSIP fosters changes in attitude and understanding of older and younger participants toward each other, as well as understand the mechanisms by which these changes occur. Misconceptions about age and gender may be challenged through (a) the participatory production phase and/or (b) the viewing of the videos produced. Our experience has been that the DST approach manages to foster more in-depth connections, communication, and meaningful dialogue and this is certainly worth exploring more empirically.

## 4. Research Project

A research and evaluation arm for PADSIP was thus set up 2021. We appointed a research manager to establish and run a pilot research study examining the (a) feasibility (completion); (b) acceptability (satisfaction); and (c) any changes and impacts on psychosocial aspects of older adults' lives, in line with the aims of the program. A pre–post design is employed with measures including the North-Western Ego Integrity Scale (NEIS-5; Janis et al. 2011), Geriatric Depression Scale (GDS-5; Hoyl et al. 1999), Three-item Loneliness Scale (TiLS-3; Hughes et al. 2004), Satisfaction with Life Scale (SwLS-5; Diener et al. 1985), Identity Validation Scale (IVS-5; Stargatt 2022), and Attitudes towards Ageing Questionnaire (AAQ-24; Laidlaw et al. 2007). Additionally, a purpose-built Program Satisfaction Survey and semi-structured interviews or focus groups are employed post-program. While this project is ongoing, preliminary analyses show positive trends across various outcome measures. Findings will be reported in future publications.

## 5. Internal Evaluation

Whilst the research project only focuses on the perspectives of older adult participants, input from younger participants, volunteers, and staff involved are also documented for internal program evaluation. This data provides the PADSIP team with insight into what works, what can be improved, and ideas to incorporate into the program design as well as implementation process, allowing it to continuously evolve.

## 6. Summary and Future Directions

In summary, over the past 15 years, the Positive Ageing Digital Storytelling Intergenerational Program (PADSIP) has witnessed transformative impacts and consolidated lessons on what makes an intergenerational digital storytelling program successful. The program has shown its potential in its versatility and adaptability to various contexts and stakeholders. Ultimately, PADSIP strives to build positive connections and relationships, strengthen identities, enhance community engagement, and combat ageism through transformative experiences. In the near future, the team will report the findings of PADSIP's pilot study and strive to conduct more empirical research to (a) uncover the mechanisms

of PADSIP and (b) conduct more longitudinal research to evaluate the sustained impacts of the program. It is also our hope for PADSIP to be more widely implemented and are in the process of developing a toolkit. With the emerging growth in our aged population, there is an imperative to build greater intergenerational communication, understanding, and connection.

**Author Contributions:** Conceptualization, M.S.; methodology, M.S. and L.Z.Y.L.; validation, M.S. and L.Z.Y.L.; formal analysis, L.Z.Y.L.; investigation, M.S. and L.Z.Y.L.; resources, M.S. and L.Z.Y.L.; data curation, L.Z.Y.L.; writing—original draft preparation, M.S.; writing—review and editing, L.Z.Y.L.; visualization, M.S. and L.Z.Y.L.; supervision, M.S.; project administration, M.S. and L.Z.Y.L.; funding acquisition, M.S. All authors have read and agreed to the published version of the manuscript.

**Funding:** This research was funded by the Danks Trust.

**Institutional Review Board Statement:** The study was conducted in accordance with the Declaration of Helsinki, and approved by the Human Research Ethics Committee of Swinburne University of Technology (protocol code 5779, approved on 2 July 2021).

**Informed Consent Statement:** Informed consent was obtained from all subjects involved in the research study.

**Data Availability Statement:** The data from the research study is currently not publicly available due to ongoing data collection and analysis. Further information may be available from the corresponding author on reasonable request.

**Acknowledgments:** Sunil Bhar, Therese Den Dulk, Debbie Shaw, Sean Summers, Craig McPherson, Bronwyn Billimoria, Philippe De Montignie; and the rest of our PADSIP family.

**Conflicts of Interest:** The authors declare no conflict of interest.

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
