# Peer review of "Reflections on the Impact of an Intergenerational Digital Storytelling Program on Changing Attitudes and Fostering Dialogue and Understanding across the Generations"

_socsci, doi:10.3390/socsci12110606_

Round 1
Reviewer 1 Report
Comments and Suggestions for Authors
The heading "Planning: Several Stages will be mentioned in this section" seems a little out of place. I suggest simply calling this section "Planning" and then letting the first sentence ("The planning stage involves several factors") do the rest of the work.
Under the subheading, "Training" it would be interesting to learn more about what kind of disability-related training is taking place. Given that digital storytelling is widely used in critical disability studies and autism studies, I suggest the authors expand this section to describe in detail their training efforts around accessibility, disability culture, etc. in line with DS work in the disability sphere.
The summary could be strengthened by adding a few more sentences about what is learned through the PADSIP work, and directions for the future.
Author Response
Dear Reviewer / Editing Team,
Please see the attachment.
Thank you!

Reviewer 2 Report
Comments and Suggestions for Authors
Your paper effectively underscores the transformative power of Digital Storytelling in bridging generational divides. PADSIP's holistic and inclusive approach, grounded in therapeutic practices, offers a refreshing perspective on community engagement. While you've adeptly highlighted both strengths and challenges, the program's adaptability and commitment to continuous improvement shine through. Well done!
Considering the immense potential of PADSIP, addressing logistical and administrative challenges will further elevate its impact. Streamlining volunteer roles and cementing collaborations with the right external partners could be pivotal. I look forward to seeing how the program evolves by addressing these challenges. Keep up the inspiring work!
Author Response
Dear Reviewer/ Editing Team,
Please see the attachment.
Thank you!

Reviewer 3 Report
Comments and Suggestions for Authors
In this article, the author(s) offer interesting ideas about their experiences with digital storytelling within an intergenerational program. As I read the article, I was left wondering: What is the main focus of the article? The author(s) outlined the purpose of the article in the (1) abstract and (2) at the beginning of the second paragraph.
(1) "The article underscores the transformative impact of the program, both on personal levels and within the broader school community, aged care environments, and society as a whole."
(2) "This article offers a reflection on an intergenerational Digital Storytelling Program called “Positive Ageing Digital Storytelling Intergenerational Program (PADSIP)” which started in Melbourne, Australia in 2007."
Please, clearly state the purpose of the article and organize the outline accordingly. Readers will be grateful for a clear overview and structured presentation.
More citations and references are needed. For example, at the end of the first paragraph, the author(s) write "The understanding across generations that such programs can generate can also go a long way in finding creative solutions to the issues that we all face today, especially in the area of Social Justice." Here a broad idea is introduced about the need for creative solutions to respond to inequities in society. However, the author(s) fail to ground this idea in literature . As reader, I am left wondering what specific social justice issues the program facilitators want to address. How have previous researchers linked IG programs to social justice?
The titles and subtitles in the article seem rather disorganized. For example, the fourth paragraph seems to outline rationales for using digital storytelling within an IG setting. Maybe the two components of digital storytelling and IG programming should be discussed as part of the literature review. Then, in a program description, the author(s) can outline the roles persons played in implementing the program and how participants engaged in storytelling.
Theoretical foundations or frameworks are good to outline. They inform the measurements that evaluators can use to determine outcomes of the program. For each theory and practice the author(s) mentioned, there needs to be at least one citation and reference. Who are prior researchers and how have they applied the theoretical frameworks to IG work?
In the planning section, important points are made about the focus and engagement of persons, planning of activities, and logistical considerations. It would be helpful to highlight each of the points with specific examples. Readers have to be able to paint a picture of the planning process and components. How are the discussed points stages?
What exactly are the outcomes that the facilitators and researchers targeted? Are the reduction of loneliness and isolation the major factors the author(s) evaluated or plan to evaluate? Or are the outcomes as listed in the title the key drivers of the program (i.e. changing attitudes and fostering dialogue and understanding through meaningful conversation)?
How did evaluators of the program collect information and data to analyze findings? If no data is available yet, please explain how anecdotal information and reflections were gathered. Given the focus of the article as a general reflection, the types of reflections, the number of persons engaged in reflections, and the information gathering techniques used have to be explained.
Outcomes and results of the program should be reported in a separate section of findings.
The author needs to explain further the concepts of horizontal and vertical connections. Why are only vertical connections allowing for storytelling? Needs to be clarified and better referenced.
Overall, more citations/references and examples are needed. Readers need to be able to understand the planning and implementation procedures that the facilitators employed. Readers also need to know how facilitators and researchers used reflection on practice and for evaluating the program. Help readers to paint a picture of how facilitators planned, implemented, and reflected throughout the IG storytelling process!
Readers need to understand how the online storytelling programs were structured and implemented.
Readers need to clearly understand the different types of implementation within specific settings and with particular participants.
PADSIP appears to have broad engagement of students and elders. Highlighting the number of students and elders and the number of sessions and weeks would be valuable information to gain. A table in which numbers of participants (both generations) and facilitators and number of sessions would be very helpful. With a table, readers could gain an overview of the program's extent.
Overall, the paper will benefit from a clear organization including sections in which the author(s) describe specific aspects of the program. Maybe sections (i.e. headings and subheadings) similar to the ones suggested in the following might work:
Introduction (including a brief literature review)
Theoretical Frameworks
Program Overview
- History and Context(s)
- Participants
- Structure of program (number of sessions and timeframes, activity procedures and flow of individual sessions)
Evaluation Methods
- Data collection
- Data analysis
- Anticipated outcomes (may include early findings)
- Future research
Author(s) will be able to improve their writing by consulting the Publication Manual of the American Psychological Association (APA) 7th edition. The chapter on elements and formats might be particularly helpful.
I hope that my suggestions are helpful.
Comments on the Quality of English Language
The author's command of English is good. In a few areas, careful editing with provide clarity for readers. For example, on page 2, line 91, Did the author mean to write "corroborate" meaning to confirm with evidence?
Some sentences are long. Creating two sentences will benefit clarity of writing. For example, page 2, lines 46-48. This sentence is confusing and too long. Make at least two out of it. What are some positive outcomes that the program targets?
The Publication Manual of the American Psychological Association (APA) 7th edition has many good suggestions for improving clarity of writing.
Author Response

(The authors gave the same response as above.)

Reviewer 4 Report
Comments and Suggestions for Authors
Dear authors! I enjoyed reading your articles.
Despite the fact that in your text you describe a very important program that is really relevant in the modern world, it should be noted that your description does not correspond to the classic version of a scientific article.
In addition, it also does not meet the requirements stated on the journal's website and the special issue in which you submit your material.
The article should include the most recent and up-to-date references in this area. The structure should include Abstract, Key words, Introduction, Materials and methods, Results, Discussion.
Therefore, I would recommend that you carefully study the requirements of the journal, review articles that are close to your topic, and rework your material into the structure of a scientific article.
Author Response

(The authors gave the same response as above.)

Round 2
Reviewer 3 Report
Comments and Suggestions for Authors
The authors present a better paper overall. Primarily, the organization of the paper is clearer. As a reader, I gain a valuable overview of the PADSIP and better understand its aims and organizational structure. Some of the headings need further edits and introductory information for logical organization of thought.
For example, the concept of loneliness and isolation appears as a heading under Program Aims. It appears that the program aims of PADSIP are (lines 84-85) “......to improve the emotional, social and mental health of all participants, including mood, affect, meaning in life, self-esteem, identity, and social connectedness.”
Based on this goal statement, I expected a discussion of each of the program aims to follow. If the authors decide to only discuss the reduction of loneliness and social isolation as a targeted outcome of the PADSIP for elders and young persons, then they need to make a clear argument for their primary focus on loneliness and social isolation. If social isolation and loneliness are used as a rationale for creating intergenerational programs, then this section belongs in the introduction.
Given that the authors outline program evaluation foci and measures later in the paper, the program aims should parallel the aims targeted in the evaluation. If the discussion remains in the program aims section, then there should be more than one concept discussed under the title of Program Aims. The program evaluation section should then follow the concepts outlined under program aims.
In the evaluation section, make sure to very clearly state what early findings are standing out. The authors point to some outcomes without elaborating, which leaves the reader wondering how the broader communities benefit from the program. For example, in lines 309-310, the authors write:
“These programs have anecdotally and observationally made significant impacts, not only at a personal level for the participants, but also for the broader school community, aged care settings, and the local community.”
As a reader I am left wondering “how so”?
Under Program Elements (line 146), there should be an introductory paragraph to help the reader know what is to come in this section of the paper. I suggest a process-oriented discussion of the program elements so that readers know how teams of facilitators were recruited, how facilitators recruited elders and youth as participants, and how teams came together to reflect on the benefits and challenges of the program. Help the reader to paint a picture of the program and share examples about some of the actions that facilitators took to engage participants!
Please, review clarity of writing one more time. Some long sentences remain. For example, lines 119 to 121 state “Through telling one’s story in a trusting atmosphere and it being richly acknowledged, experiences and emotions are shared and common ground can be found which leads to a breakdown of feelings of isolation and difference; setting the scene towards a more nurturing environment.” Using active rather than passive voice and creating at least two sentences here will improve clarity.
Overall, it is necessary for the authors to revise the paper again to be certain that they do what they propose to do in the abstract "outlining its various elements, reporting observed impacts on participants and the wider community, as well as underscoring future directions."
They have improved the outlining of elements section and outlined future directions more clearly. They need to further improve the reporting of observed impacts on participants and the wider community.
I recommend rewriting again for clarity purposes.
Comments on the Quality of English Language
Please, be sure to double check long sentences. Avoid them by creating two or three clear sentences. Also, make sure to use active rather than passive voice.
Author Response
Dear Reviewer,
Please see attached document for responses.
Thank you!

Reviewer 4 Report
Comments and Suggestions for Authors
Dear authors!
Thank you for your response and editing of your article.
It should be noted that the material itself has been developed and improved.
The authors introduced and corrected many shortcomings in their text.
However, one point is questionable.
This Brief Report is dedicated to the impact of an Intergenerational Digital Storytelling Program on changing attitudes and understanding across the generations
At the same time, the vast majority of sources refer either to the last century or 15 years ago, which seems irrelevant to the topic related to digitalization.
Thus,
The article is devoted to extremely relevant and pressing issues.
However, I would advise supplementing the article with newer sources.
Author Response
Dear Reviewer,
Please see the attached document for responses.
Thank you!
